# A Novel FLCN Variant in a Suspected Birt–Hogg–Dubè Syndrome Patient

**DOI:** 10.3390/ijms241512418

**Published:** 2023-08-04

**Authors:** Erika Bandini, Valentina Zampiga, Ilaria Cangini, Mila Ravegnani, Valentina Arcangeli, Tania Rossi, Isabella Mammi, Francesca Schiavi, Stefania Zovato, Fabio Falcini, Daniele Calistri, Rita Danesi

**Affiliations:** 1Biosciences Laboratory, IRCCS Istituto Romagnolo per lo Studio dei Tumori (IRST) “Dino Amadori”, 47014 Meldola, Italy; valentina.zampiga@irst.emr.it (V.Z.); ilaria.cangini@irst.emr.it (I.C.); tania.rossi@irst.emr.it (T.R.); daniele.calistri@irst.emr.it (D.C.); 2Romagna Cancer Registry, IRCCS Istituto Romagnolo per lo Studio dei Tumori (IRST) “Dino Amadori”, 47014 Meldola, Italy; mila.ravegnani@irst.emr.it (M.R.); valentina.arcangeli@irst.emr.it (V.A.); fabio.falcini@irst.emr.it (F.F.); rita.danesi@irst.emr.it (R.D.); 3Familial Cancer Unit, Veneto Institute of Oncology IOV IRCSS, 35128 Padova, Italy; isabella.mammi@iov.veneto.it (I.M.); francesca.schiavi@iov.veneto.it (F.S.); stefania.zovato@iov.veneto.it (S.Z.)

**Keywords:** case report, genetic testing, FLCN, NGS, genetic variants

## Abstract

Subjects with pathogenic (PV) and likely pathogenic (LPV) FLCN variants have an increased risk of manifesting benign and malignant disorders that are related to Birt–Hogg–Dubé syndrome (BHDS): an autosomal dominantly inherited disorder whose severity can vary significantly. Renal cell carcinoma (RCC) development in BHD (Birt–Hogg–Dubé) patients has a very high incidence; thus, identifying this rare syndrome at early stages and preventing metastatic spread is crucial. Over the last decade, the advancement of Next Generation Sequencing (NGS) and the implementation of multigene panels for hereditary cancer syndromes (HCS) have led to a subsequent focus on additional genes and variants, including those of uncertain significance (VUS). Here, we describe a novel FLCN variant observed in a subject manifesting disorders that were suspected to be related to BHDS and with a family history of multiple cancers.

## 1. Introduction

Originally described in 1977, BHDS is an autosomal dominant disorder that is manifested by the development of benign cutaneous lesions, particularly fibrofolliculomas. Affected subjects also have a high risk of developing pulmonary cysts with associated spontaneous pneumothoraces, benign renal cysts, RCC, and other rare diseases (lipomas, parathyroid adenomas, parotid gland tumors, and colonic polyps/tumors) [1]. These clinical features are caused by germline mutations in the folliculin FLCN gene, a tumor suppressor which consists of 14 exons located on chromosome 17p11.23 [2]. Patients with PV and LPV FLCN variants have an increased risk of manifesting benign and malignant kidney disorders. Bilateral, multifocal renal lesions are common and can be histologically differentiated, mainly showing hybrid oncocytic masses, chromophobe RCC and clear cell renal carcinoma (ccRCC). It has been predicted that 12% to 34% of BHD individuals develop kidney cancers with effective diagnosis at a mean age of 50.7 y.o. and earliest diagnosis at age 20 [3]. While early-stage partial nephrectomy is the first opportunity for an FLCN-deficient RCC treatment, the advanced BHDS-associated RCC is still demanding to treat due to a lack of defined therapeutic targets [4]. To date, more than 200 alterations in the FLCN gene have been identified and reported in the FLCN Leiden Open Variation Database. The most usually detected variants concern protein truncations, including frameshift (small deletions or insertions), nonsense, or splice variants. The detection rate of FLCN mutations in subjects with BHDS features is almost 90% [5]. It has been observed that FLCN mutations in exon 9 are associated with an increased number of lung cysts, and exon 9 and 12 alterations are correlated with more episodes of spontaneous pneumothorax [6]. The identification of gene mutation carriers in relatives of HCS families has important implications for cancer prevention, early diagnosis, and clinical decision-making. In order to deal with these high-risk individuals, clinical practice guidelines and specific genetic counseling programs have been integrated into the context of healthcare institutions [7]. There may be several families with BHDS, and unfortunately, the syndrome is likely to be underdiagnosed. As it is a rare disease, diagnosis, and management represent a crucial point. Here, we present the case of a subject manifesting disorders suspected to be related to BHDS and report an identified FLCN variant of uncertain significance never described before.

### Case Presentation

A 47-year-old patient was recruited by our Oncogenetic Counselling Service after the request of the Oncology Unit, following a history of an RCC at 46 y.o. of hybrid histotype oncocytic/chromophobic. The patient underwent a right renal nephrectomy in January 2021 and subsequent peritoneal lymphadenectomy. The histological analysis of neoplastic cells was found to be positive for the expression of PAX-8 and CD117, and cytokeratin 7. He referred to following a vegetarian diet and did not smoke. He reported allergies to cefazoline sodium. Over the year following surgery, the patient experienced secondary disorders, retroperitoneal adenopathies, retrocaval lymph nodes, and secondary malignant tumors of the bone and bone marrow. Due to the nature of the carcinoma, he was treated with Pembrolizumab and Axitinib in November 2021, in addition to Zometa. In September 2022, he underwent second-line therapy with Cabozantinib, being affected by stage IV RCC and secondary diseases. To date, due to a worsening state of health correlated to hepatic, pulmonary, and abdominal secondary diseases, the patient is still on a Cabozantinib regimen. During the genetic counseling, the patient reported a childhood episode of spontaneous pneumothorax (PNX) that led to hospitalization, which the geneticist evaluated as a possible correlation factor to BHDS. The patient did not report any other noteworthy pathologies and no first-degree family history of cancer-related diseases, except for the sister who developed melanoma at the age of 23, followed by thyroid cancer at 42 y.o. with a relapse at 43 y.o. Since the patient is considered to be at family risk of developing melanoma, he is periodically followed up by the dermatology unit, although to date, he has not developed related symptoms, except palmar-plantar erythrodysaesthesia, which, however, was probably due to therapy toxicity. After collecting family history information, a genetic test was required and performed by the Genetics Unit in order to find a possible association with BHDS, in accordance with the diagnostic guidelines proposed by the European Consortium, which require 1 major or 2 minor criteria [8]. The pedigree of the family members included in the study is shown in Figure 1. Furthermore, the patient had a daughter and a son, but being a minor, the genetic test is not allowed to be carried out until the age of 18 y.o.

## 2. Results

Our genetic test revealed a novel and previously not reported missense variant in the FLCN gene. The analysis highlighted the presence of heterozygosity of the variant FLCN c.1610 G>A, p. (Ser537Asn), which is classified as VUS (class 3). This variant is absent in the main population databases and is not reported in a specific locus database LOVD FLCN nor in ClinVar, as well as has never been described before. This VUS presents a sequence characterized by a replacement of the serine with an asparagine at codon 537 of the FLCN protein. Based on currently available references and following the recommendations of the American College of Medical Genetics and Genomics (ACMG) [9], the variant c.1610 G>A, p.(Ser537Asn) of FLCN has insufficient and conflicting evidence. The sister was found negative for the presence of the FLCN variant, and since she had melanoma, she was also tested for the CDKN2A and CDK4 resulting wild type (WT). It is not possible to know the carrier status of the son and the daughter, as they are still minors, and it is not yet possible to perform the genetic test. Furthermore, it would have been very useful to establish the parental origin of the variant, but we are currently unable to perform the test since both parents are close to 90 y.o and are suffering from senile disorders.

## 3. Discussion

According to the ClinVar database, most disease-linked FLCN variants in the coding regions are frameshift mutations leading to C-terminal truncations of the FLCN protein, and nearly 150 PVs have been identified, among which the most frequent pattern is a frameshift mutation within a coding exon [10,11]. Carriers of FLCN gene mutations may be asymptomatic or show different degrees of cutaneous, pulmonary, or renal features, as well as the fact that BHDS can arise regardless of gender and age, although it tends to emerge in the third or fourth decade of life. Normally, affected subjects exhibit skin lesions and spontaneous recurrent pneumothoraces due to pulmonary cysts, but sometimes BHDS can be associated with pulmonary cysts and/or relapsing pneumothoraces in the absence of any cutaneous or renal involvement [12]. According to studies conducted in Western countries, 1–12% of the subjects with FLCN germline mutations manifested renal tumor as the only sign of BHDS, whereas additional studies in different countries reported no one with the disease having a renal tumor [13]. Several genetic tumor disorders are well known for their association with increased cancer risks, while in others, including BHDS, such an association is still a matter of discussion. Furthermore, there is a lack of clear and accurate guidelines regarding the diagnosis and management of the disorder. All scientific reports describe the heterogeneity of this syndrome which represents a real difficulty in patient management as the diagnosis is often delayed for years [14]. In this study, we report on an observed c.1610 G>A, p.(Ser537Asn) VUS FLCN variant never described before, which was identified in a patient with suspected BHDS, in accordance with the recent European BHDS Consortium Criteria [15]. In light of the data obtained in the proband, it was considered that a targeted analysis should be carried out on the sister, who had melanoma and thyroid cancer history, to evaluate her possible status as a carrier; however, she was negative for the observed FLCN variant. The identification of this novel missense variant of FLCN could expand the mutational spectrum of the FLCN gene in patients who are at risk of developing BHDS. To date, the newly identified variant c.1610 G>A p. (Ser537Asn) is absent in the main population databases, and it has never been reported in previous studies. Since the patient reported an episode of pneumothorax in childhood and a RCC history, together with the presence of a FLCN variant, it could be conceivable to assume a suspected BHDS. Previous studies describe a correlation between a history of pneumothorax and the syndrome, reporting a percentage of patients carrying FLCN gene mutations and a past history of pneumothorax [16]. As clinical manifestations can differ according to age group and patient features, BHDS remains difficult to diagnose, likely resulting in many cases of missed diagnosis. Indeed, Menko et al. proposed modified diagnostic criteria based on the detection of DNA mutations in FLCN, citing other minor criteria. It is worth mentioning a study that described the presence of an uncertain FLCN variant (VUS; c.1333 G>A) in a male patient diagnosed with RCC and again with a FLCN VUS (c.384 C>G) in a 67-year-old female with histological features suggestive of BHDS [17]. Overall, few studies have investigated the role of VUS in this disorder [10,18,19], while it would be useful to spread the observations of these variants to broaden the spectrum of mutations involved. Different ethnic groups must also be taken into consideration, as some studies have demonstrated the variability of the FLCN mutation spectrum among different groups, for instance, Caucasians, with more skin and kidney lesions and fewer lung cysts and pneumothorax [19]. The encouraging possibilities of personalized medicine have opened in the direction of routine broad-scale sequencing and have increased the importance of genetic counseling for hereditary cancers. Nowadays, the improvements in NGS and targeted therapy have enriched the clinical approach and management of patients affected by genetic disorders and neoplastic diseases [20]. Observations on FLCN novel variants that might cause renal and further symptoms when FLCN is mutated in individuals can open up new research directions on the development of diagnosis and effective targeted therapies for BHD patients [21]. A recent study reported that the prevalence of BHDS is expected to be far higher than assumed so far and that only a minority of subjects with a phenotype correlated to the disease were actually diagnosed. It becomes equally necessary to revise the guidelines for genetic testing on the basis of novel insights on the prevalence and the penetrance of hereditary disorders since, until now, genetic testing indications have still mainly been based on the most serious clinical manifestations, precluding a genetic testing opportunity in patients with a mild phenotype, resulting in many remaining underdiagnosed [22]. New models of care are needed to ensure the implementation of surveillance guidelines for BHDS. Individuals with a suspected genetic susceptibility to BHD must be initially referred for a clinical assessment by a geneticist, along with at-risk family members who should be offered predictive testing when expected. These managements could be implemented with genetic insights that might reveal the presence of novel and unusual variants in order to increase surveillance and facilitate for geneticists rare syndromes diagnosis.

## 4. Materials and Methods

### 4.1. Patient Collection

Involved subjects were referred to the genetic counseling of the Genetics Unit of IRST IRCCS D.Amadori and were enrolled and included in the study in 2021. The study was approved by the institutional review board (Ethics Committee IRST IRCCS-AVR, 2207/2012) and conducted in accordance with the Declaration of Helsinki. Written informed consent was obtained from the subjects before this study. Information about possible tumors and malignancies related to family history of first- and second-degree relatives was also collected.

### 4.2. Blood Collection and DNA Extraction

Peripheral blood samples were collected and stored at −80 °C at the Biosciences Laboratory of the IRCCS Istituto Romagnolo per lo Studio dei Tumori “Dino Amadori.” Genomic DNA was extracted using the Maxwell RSC Whole Blood DNA Kit (Promega, Italy) according to the manufacturer’s instructions. DNA was quantified by a Qubit fluorometer (Thermo Fisher Scientific, Waltham, MA, USA) through the Qubit dsDNA BR Assay Kit (Thermo Fisher Scientific, Waltham, MA, USA).

### 4.3. Next-Generation Sequencing

For NGS, analysis was performed through two panels containing targeted genes: a Custom Panel was validated in-house for the Endocrine and Renal Tumors kit REK_IOVP v4, which included 52 genes, as shown in Table 1, and the enrichment protocol of SOPHiA Custom Hereditary Cancer Solution (CHCS) v1.1 by SOPHiA GENETICS (Saint-Sulpice, Switzerland), which investigates 32 cancer predisposition genes, as described in Appendix A. For the Renal Custom Panel, sequencing libraries were prepared from 200 ng of genomic DNA through the QiaSeq Fx kit (Qiagen). Regions of interest were enriched for each library with xGen Custom Hyb Panels (Endocrine and Renal Tumors kit REK_IOVP v4) and xGen Hybridization Capture Core Reagents (IDT) processed on a Hamilton NGS Microlab STARlet, using experimental parameters according to the manufacturer’s instructions. The sequencing of NGS libraries was performed through a MiSeq (Illumina) paired-end 2 × 301-bp DNA sequencing platform with a MiSeq Reagent Kit v3 (600-cycle, Illumina), according to the manufacturer’s procedure. The analysis of gene variants, including the FLCN and copy number variations (CNVs), was performed through the SOPHiA DDM™ Platform (Sophia Genetics), and an interpretation was carried out according to the American College of Medical Genetics and Genomics (ACMG) and the Association for Molecular Pathology (AMP) Standards and Guidelines [9], along with the most recent Sequence Variant Interpretation ClinGen specifications-General Recommendations for Using ACMG/AMP Criteria (https://clinicalgenome.org/working-groups/sequence-variant-interpretation/). For the CHCS panel, sequencing libraries were created from 200 ng of genomic DNA and were processed on a Hamilton NGS Microlab STARlet, according to the manufacturer’s instructions. The sequencing was performed using the MiSeq sequencer platform (Illumina) and MiSeq Reagent Kit v3 600 cycles, configured in 2 × 151 cycles in accordance with the manufacturer’s instructions. Data output files (FASTQ) were uploaded on the SOPHiA DDM Platform v5.5.0 (SOPHiA GENETICS, Saint-Sulpice, Switzerland) for additional analysis. The classification of variants was performed through the main mutation databases and tool prediction software, Leiden Open Variation Database (LOVD), ClinVar, dbSNP, and Varsome, and were reported according to their available clinical interpretation. Variants automatically annotated by the platform were manually checked on the main human genomic databases. Furthermore, in order to define the connections between the proteins encoded by the genes included within the custom panel, a protein–protein interaction (PPI) network was prepared using Cytoscape tool v3.10.0 [23], as illustrated in Appendix A.

## Figures and Tables

**Figure 1 ijms-24-12418-f001:**
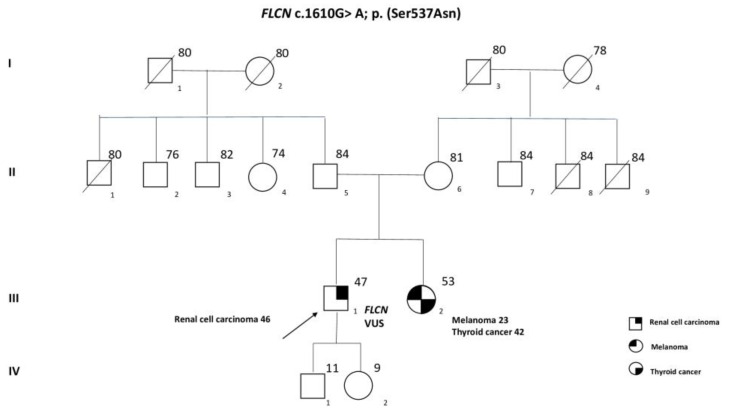
Family pedigree. Circles represent females and squares represent males. I–IV represent generations. Symbols with a quarter represent cancer patients. Symbols with a slash indicate deceased individuals. The arrow points to the proband.

**Table 1 ijms-24-12418-t001:** Panel of 52 genes used for NGS analysis through Custom Panel for Endocrine and Renal Tumors REK_IOVP_v4.

AIP	AP2S1	ARMC5	ATRX	BAP1	BRK1	CASR	CDC73	CDKN1A	CDKN1B
CDKN2B	CDKN2C	DNMT3A	EGLN1	EGLN2	EPAS1	ESR2	FGFR1	FH	FLCN
GCM2	GNA11	H3F3A	KIF1B	KMT2D	MAX	MDH2	MEN1	MERTK	MET
NF1	PBRM1	PDE11A	PDE8B	PRKACA	PRKAR1A	PTEN	RET	SDHA	SDHAF1
SDHAF2	SDHAF3	SDHAF4	SDHB	SDHC	SDHD	SLC25A11	TMEM127	Tp53	TSC1
TSC2	VHL								

## Data Availability

The data presented in this study are available on request from the corresponding author. The datasets presented in this article are not readily available because they are part of a genetic data obtained from the analysis of patients.

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
