# Peer review of "A Novel FLCN Variant in a Suspected Birt–Hogg–Dubè Syndrome Patient"

_ijms, 2023, doi:10.3390/ijms241512418_

Round 1

Reviewer 1 Report

The study of Dr. Bandini et al. provided a case report on a novel FLCN variant which may associated with BHDS. The report may provide a new candidate in this expertise.

There are a few issues the authors should address:

1. To test the presence of the variant, authors may consider also the sequencing on the parents of the patient. This may provide an information if the variant is inherited from his parents. 

2. Are there any other known pathogenic variants present in this patients? Since this variant classified as VUS, more information included would be better for future study on its effects.

Author Response

  1. To test the presence of the variant, authors may consider also the sequencing on the parents of the patient. This may provide an information if the variant is inherited from his parents. 

Answer: We would like to thank the Reviewer for the insightful suggestions and for appreciating our attempt to describe this variant with the aim of making it helpful for studies of cancer predispositions. Actually it may also be a de novo variant, but unfortunately the parents are very old and close to 90 y.o. One of them is also affected by senile dementia, so we are currently unable to perform the test. Thanks to this valuable comment, however, we have added this observation in the paragraph of the results. Furthermore, we will plan to test the son and daughter as soon as they are > 18 y.o.

  1. Are there any other known pathogenic variants present in this patients? Since this variant classified as VUS, more information included would be better for future study on its effects.

Answer: No, there are no pathogenic variants in either the patient or the sister. For this reason we would like to highlight the uncertain variant to report it as a non-canonical variant potentially involved in this rare syndrome.

Reviewer 2 Report

some pitfalls require enhancement as follows:

1. in abstract and introduction, clarify what BHD stands for?

2. Revise your abbreviations, you must clarify the full name for the first time and repeat the abbreviation next.

3. You stated that the history of the patient for RCC started when he was 46 Y.O. i.e. last year (2022), why your registration of the study in 2012??

4. you have to add some demographic data and related statistics for the collected population.

5. I suggest performing PPI network analysis for the 52 genes used for NGS, maybe by cytoscape software.

should be improved

Author Response

  1. in abstract and introduction, clarify what BHD stands for?

Answer: We thank the Reviewer for the suggestion. We added this omission in the abstract and modified BHDS in the introduction and in the rest of the manuscript.

  1. Revise your abbreviations, you must clarify the full name for the first time and repeat the abbreviation next.

Answer: We thank the Reviewer for the suggestion. We have limited the use of abbreviations in the text.

  1. You stated that the history of the patient for RCC started when he was 46 Y.O. i.e. last year (2022), why your registration of the study in 2012??

Answer: we thank the Reviewer for the comment. Since we are a Molecular Diagnostic Unit and we do not routinely recruit patients for research purpose, in our Institute we had to establish specific guidelines for the enrollment of patients and to involve them in research studies. Thus, being the patients routinely enrolled and signing the same informed consent for the same genes and panels, a single study was created for permission to collect blood samples from 2012 onwards.

  1. you have to add some demographic data and related statistics for the collected population.

Answer: we thank the Reviewer for the comment. We are aware that a statistical study would actually be very interesting and the manuscript would certainly acquire more value. However, the intent of the work was to make an observational study. Since BHDS  is a rare syndrome, we do not have sufficient data for a population study. Being a case report and having a limited study population, it is not possible, also in agreement with our statistician, to carry out a statistical analysis. But we thank the Reviewer for the suggestion, and we intend to expand the study if we have other future data to integrate for the analysis of further FLCN variants and BHD cases.

  1. I suggest performing PPI network analysis for the 52 genes used for NGS, maybe by cytoscape software.

Answer: we thank the Reviewer for the comment. We integrated a PPI network analysis in a supplementary image and also with a reference.

Round 2

Reviewer 2 Report

we would to thank the authors for their responses